# Early-Stage Identification of Powdery Mildew Levels for Cucurbit Plants in Open-Field Conditions Based on Texture Descriptors

Claudia Angélica Rivera-Romero [1], Elvia Ruth Palacios-Hernández [2], Osbaldo Vite-Chávez [1] and Iván Alfonso Reyes-Portillo [3,*]

[1] Unidad Académica de Ingeniería Eléctrica Plantel Jalpa, Universidad Autónoma de Zacatecas, Jalpa 99601, Mexico; c.a.riveraromero@uaz.edu.mx (C.A.R.-R.); osvichz@uaz.edu.mx (O.V.-C.)

[2] Facultad de Ciencias, Universidad Autónoma de San Luis Potosí, San Luis Potosí 78295, Mexico; epalacios@fciencias.uaslp.mx

[3] Academia de Ingeniería en Sistemas y Tecnologías Insdustriales, Universidad Politécnica de San Luis Potosí, San Luis Potosí 78295, Mexico

[*] Correspondence: ivan.reyes@upslp.edu.mx; Tel.: +52-294-121-45-77

**Abstract:** Constant monitoring is necessary for powdery mildew prevention in field crops because, as a fungal disease, it modifies the green pigments of the leaves and is responsible for production losses. Therefore, there is a need for solutions that assure early disease detection to realize proactive control and management of the disease. The methodology currently used for the identification of powdery mildew disease uses RGB leaf images to detect damage levels. In the early stage of the disease, no symptoms are visible, but this is a point at which the disease can be controlled before the symptoms appear. This study proposes the implementation of a support vector machine to identify powdery mildew on cucurbit plants using RGB images and color transformations. First, we use an image dataset that provides photos covering five growing seasons in different locations and under natural light conditions. Twenty-two texture descriptors using the gray-level co-occurrence matrix result are calculated as the main features. The proposed damage levels are 'healthy leaves', 'leaves in the fungal germination phase', 'leaves with first symptoms', and 'diseased leaves'. The implementation reveals that the accuracy in the L * a * b color space is higher than that when using the combined components, with an accuracy value of 94% and kappa Cohen of 0.7638.

**Keywords:** statistical tests; feature extraction; feature selection; classification; confusion matrix; accuracy

## 1. Introduction

Agriculture is one of the primary resources and involves a large community of plants as various crops with different environmental conditions. A large part of a country's economy involves the export of agricultural products that are sold daily for human consumption. The diagnosis and prevention of pathologies in crops are required tasks in agriculture. The excessive use of pesticides, inappropriate farming practices, and the abandonment of plant-disease-infected regions are causing agricultural losses. In addition, farmers confront several problems every day, such as fungal plant diseases. Different plants are highly susceptible to damage by fungi. In the case of cucurbits, there are limited studies about the damage caused by diseases and pests. According to previous studies, fungal infections such as powdery mildew (PM) begin with spore germination [1–3]. This fungus is the most common type of disease found in open-field crops.

Various techniques involving mathematical and computational processes using information obtained by digital images can be used for disease detection. Currently, some methods use images for disease and pest classification under conditions of infected plants [4–6]. Plant disease detection using on image-processing technologies generally involves a methodology that includes plant disease image acquisition, image processing, image segmentation through the region of interest, feature extraction and selection, and the

application to disease detection. An image provides sufficient information to identify characteristics that describe the severity and stage of the disease [7] because the leaves of the plant are the first organ that shows symptoms of a disease. Methods such as classification algorithms have been developed over the years for disease identification. These algorithms consist of feature extraction from images of plants experiencing problems at different disease stages.

There is a need for early-stage identification in general in diseased plants when the first symptoms appear in order to take effective control measures against a fungus. This paper proposes a method for early PM detection in cucurbit leaves based on digital images according to predefined PM damage levels. Then, the problem becomes training a set of classifiers. The first stage consists of training the classifier to distinguish between two PM damage levels. The second stage involves a voting scheme that determiens the PM damage level.

In the literature, some methodologies for detection applied to fungal diseases have been proposed. However, there are still open problems and unsolved issues related to the classification of powdery infection and prevention, such as the early detection of powdery mildew in cucurbit leaves. This scenario comprises the detection of the first symptoms of initial powdery mildew germination, which is a crucial phase for implementing management strategies to achieve eradication. Some studies only identified the disease when the plants have symptoms. However, the real problem is cases in which the plants have not yet shown symptoms. From this perspective, the innovations of the proposed methodology include (i) early symptom detection in natural conditions of crops with fungal disease; (ii) a feature extraction process in which the transformed and processed images are feature descriptors of, for example, the texture in an image; (iii) statistical feature selection is executed with the feature data to reduce the number of color components and descriptors; (iv) a nondestructive methodology for crop plants; (v) sample images are in natural lighting conditions; and (vi) disease detection and classification of powdery mildew infection in cucurbit leaves considering three phases of damage: the germination stage, the first symptoms, and when the leaves have the fungus.

*Literature Review*

Kumar et al. [8] introduced a novel exponential spider monkey optimization method to fix the significant features from a set of features generated using a subtractive pixel adjacency model. Through a support vector machine, plants were classified as diseased or healthy. The selected features for the spider monkey optimization increased the classification reliability to an accuracy of 92.12% with 82 selected features. A hybrid prediction model was developed by Lamba et al. [9] for predicting various levels of severity of blast disease using diseased plant images. This work was based on the percentage of leaf area affected by the disease. The features were extracted from an image dataset with a convolutional neural network approach. The classification accuracy of the severity level of blast disease was 97%. Kaya et al. [10] proposed a novel approach based on deep learning for plant disease detection by fusing RGB and segmented images. They considered two images as the input to a multiheaded dense-net-based architecture. They used the Plant Village database with 38 classes. The accuracy was 98.17%. Xu et al. [11] proposed a vision system with an integrated reflection–transmission image acquisition module, human–computer interaction module, and power supply module for rapid Huanglongbing (HLB) detection in the field. With six classes of identification (healthy; HLB pre-symptomatic; zinc, magnesium, or boron deficiency; or HLB-positive), a step-by-step classification model with four steps was used. The results showed that the model had an accuracy of 96.92% for all categories of samples and 98.08% for multiple types of HLB identification.

Sabat-Tomala et al. [12] used support vector machine and random forest as two machine learning algorithms to discriminate *Solidago* spp., *Calamagrostis epigejos*, and *Rubus* spp. using hyperspectral aerial images. Kasinathan et al. [13] proposed a method of insect detection based on morphological features. The classification used nine to twenty-four

insect classes using shape features and machine learning techniques. The machine learning models applied for the comparison were support vector machine, K-nearest neighbors, artificial neural network, naïve Bayesian model, and convolutional neural network (CNN). The algorithm consisted of foreground extraction and insect contour detection.

Recently, Yag et al. [14] used a new hybrid plant leaf difference as a classification model, including a flower pollination algorithm, a support vector machine, and a convolutional neural classifier. The two-dimensional discrete wavelet transform used image datasets from apple, grape, and tomato plants for feature extraction. Fernandez et al. [15] conducted a study to find the spectral changes caused by the downy mildew pathogen Podosphaera xanthii on cucumber leaves. They adapted principal component analysis to the spectral characteristics of healthy and diseased leaves. The authors used a linear support vector machine classifier, achieving an accuracy of 95%.

## 2. Materials and Methods

The general scheme, which involves a machine learning approach for the early detection of PM damage, is shown in Figure 1. Image acquisition and preprocessing are the first steps, which are followed by a feature extraction process through texture descriptors. The application selects the optimal features based on a comparison test. Binary classifiers achieve multiclassification in combination with a voting scheme and SVM blocks. Finally, the performance is evaluated with parameters that determine the optimal classification of PM damage level in cucurbit leaves.

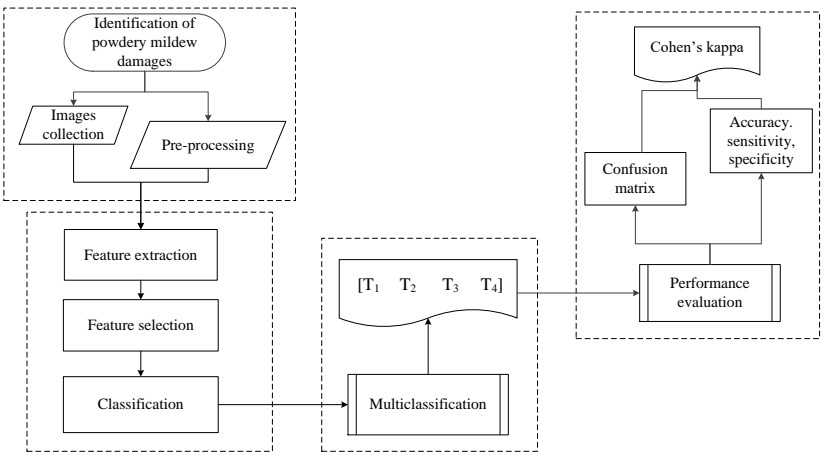

**Figure 1.** Proposed methodology for PM damage level detection, where image collection is used for feature extraction and selection. A multiclassification is operated with the results of the classification process. In the end, a performance evaluation is conducted to verify the optimal classification.

### 2.1. Acquisition

We used an image database of cucurbit plants and leaves consisting of six distinct crops in diverse locations and natural conditions: San Luis Potosí (San Luis Potosí), Jalpa (Zacatecas) and Yuriria (Guanajuato) in Mexico. During September–December 2015 ($21°69'42.1''$ N, $102°97'34.5''$ and $20°08'08''$ N, $101°01'52''$), January–April 2016 ($21°65'89.7''$ N, $102°96'80.6''$ and $21°69'43''$ N, $102°97'09.5''$), September—November 2016 ($20°21'92.7''$ N; $101°10'11.6''$ W), and March–June 2017 $21°59'75.7''$ N, $103°01'52.3''$), a sampling process was used to record the phenological data of the plants.

In open-field crops, irrigation systems were used, and preventive treatments were applied every three days for leafminers, whiteflies, spider mites, downy mildew, and powdery mildew. At the same frequency, leaves were imaged during the growing season in the morning under natural field crop conditions. The database consists of 51,260 images. Each leaf was sampled from the unfolded stage on the main stem to senescence, from the 1st true leaf to the 21st leaf of each plant. During sampling days, $D_1$–$D_{19}$ image samples

were collected. The camera of a mobile device was fixed on a plastic structure with a distance of 20 cm between the blade and the device, with a resolution of 2448 × 3264 with a 13 megapixels resolution in Joint Photographic Experts Group (JPEG format) and RGB color space (red, blue and green).

### 2.2. Proposed Powdery Mildew Damage Levels

Different plants are susceptible to fungal damage. In the case of cucurbit plants, there are limited studies on disease and pest damage. According to previous studies, fungal infections such as PM start with the germination of spores [1–3]. This disease has a spore germination cycle. The fungus appears on mature leaves when the plant is in the flowering and fruit development stages. The spore germination stage occurs when the infection structure is being formed. This process occurs over three to seven days before the first symptom becomes visible on the leaf surface. Some changes in the spore germination cycle at phenological stages $S_1$ to $S_8$ on sampling days $D_1$ to $D_{19}$ are considered as basic information for detecting different levels of leaf damage.

In Rivera-Romero et al. [16], a statistical analysis was conducted for determining a timeline (Figure 2) with sampling days and phenological growth stages with the visual assessment of PM signs and symptoms. At $T_1$, leaf development ($S_1$), lateral bud formation ($S_2$), and inflorescence emergence ($S_5$) during the first nine sampling days ($D_1$–$D_9$) are considered. From $D_{10}$ to $D_{12}$, between the main stages of flowering and fruit development ($S_6$ and $S_7$), leaves with damage level $T_2$ are monitored. Leaves with damage level $T_3$ are found in the main stages of fruit formation ($S_7$) between $D_{13}$ and $D_{16}$. The main fruit and seed growth and ripening stages are shown in $S_7$ and $S_8$ and in leaves at $T_4$ from $D_{17}$ to $D_{19}$ days of sampling. Four levels of PM damage are then defined (Figure 3): $T_1$ for healthy leaves, $T_2$ for leaves with germinating spores, $T_3$ for leaves with early symptoms, and $T_4$ for diseased leaves.

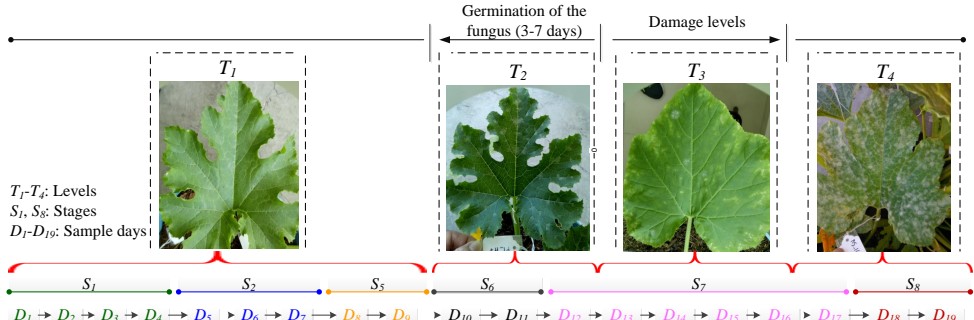

**Figure 2.** A timeline of the sampling days and the phenological growth stages to identify PM damage levels. The phenological stages ($S_1$ to $S_8$) and the sampling days ($D_1$ to $D_{19}$) are considered as basic information. Then, four PM damage levels are defined: $T_1$ for healthy leaves, $T_2$ for leaves with spore in germination, $T_3$ for leaves with the first symptoms, and $T_4$ for diseased leaves.

Because cucurbit leaves have five lobes, a region of interest (ROI) was selected for analysis. The leaves were divided into six sections ($R_1$–$R_6$) to investigate where the first symptoms were visible. A total of 465 leaves were then selected, of which 284 had first symptoms in the same region. Figure 4 shows the leaf division, where regions $R_3$ and $R_4$ have a higher incidence of first symptom appearance. This ROI selection is in agreement with the knowledge of local farmers, who confirmed that the first symptoms appear in the central upper lobe ($R_4$) of basal and mature leaves at the flowering stage.

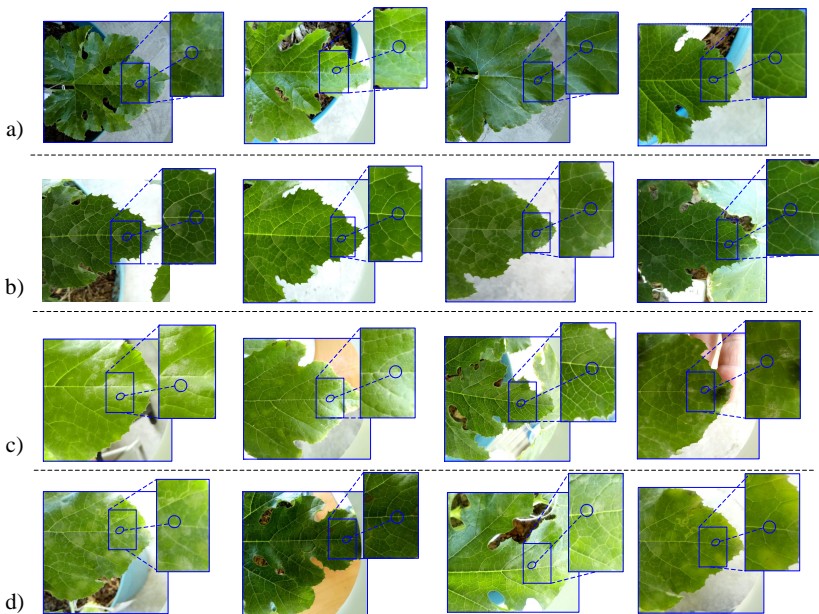

**Figure 3.** Visual evaluation of cucurbit leaves where four PM damage levels were defined: (**a**) $T_1$: healthy leaves, (**b**) $T_2$: leaves with spore in germination, (**c**) $T_3$: leaves with the first symptoms, and (**d**) $T_4$: diseased leaves.

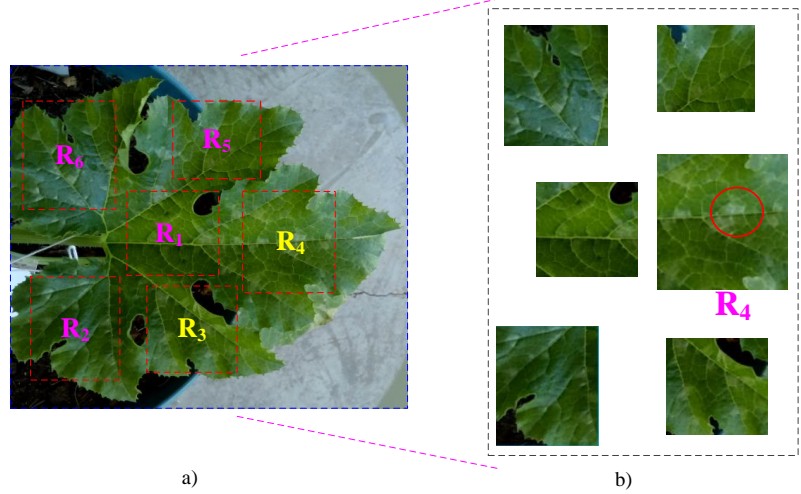

**Figure 4.** Exploration by parts of the leaf for the selection of the region of interest (ROI): (**a**) division of the leaf, central part ($R_1$), lower right lobe ($R_2$), upper right lobe ($R_3$), upper central lobe ($R_4$), upper left lobe ($R_5$) and lower left lobe ($R_6$), (**b**) first symptoms at $R_4$.

### 2.3. Preprocessing

Images were divided into four sets according to the assessed level of damage. Image samples were defined as $I(x, y)$, where $x$ represents the number of rows, and $y$ is the number of columns of a matrix, as shown in the ROI. The ROI image was defined as $R(s, t)$, where $s$ are the rows, and $t$ are the columns of a matrix that comprise the cropped image. All these images correspond to the $R_4$ region in the red, green, and blue (RGB) color space with a size of $200 \times 200$ megapixels. The ROI image dataset consisted of 5906 samples: 3610 samples were used for the damage level in $T_1$, 760 samples were used for $T_2$, 734 samples were used for $T_3$, and $T_4$ consisted of 802 samples. A contrast setting $C(p, q)$, where $p$ is the row, and $q$ is the columns, in a matrix in the range of [0.4–0.7] was employed to enhance the highlighting, followed by a spatial color transformation to different color spaces $T(s, t)$, where $s$ is the number of rows, and $t$ is the number of columns

in a matrix. Then, a color transformation was applied to the RGB color space samples into different color spaces, separating each sample into all color components (CCs): gray levels ($G(i, j)$), HSV (hue (H), saturation (S), and value (V)), $H(i, j)$), L * a * b (luminance (L), chrominance * a (A), chrominance * b (B), $L(i, j)$), and YCbCr (luma component (Y), chroma blue difference (Cb), and chroma red difference (Cr), $Y(i, j)$), where *i* is the number of rows and *j* represents the number of columns in a matrix. Thirteen processed images were extracted. Each processed sample of the different spatial colors was separated and labeled into all og its CCs. With the color space transformation, the total dataset contained 76,778 samples. Figure 5 shows the different color spaces of an ROI image.

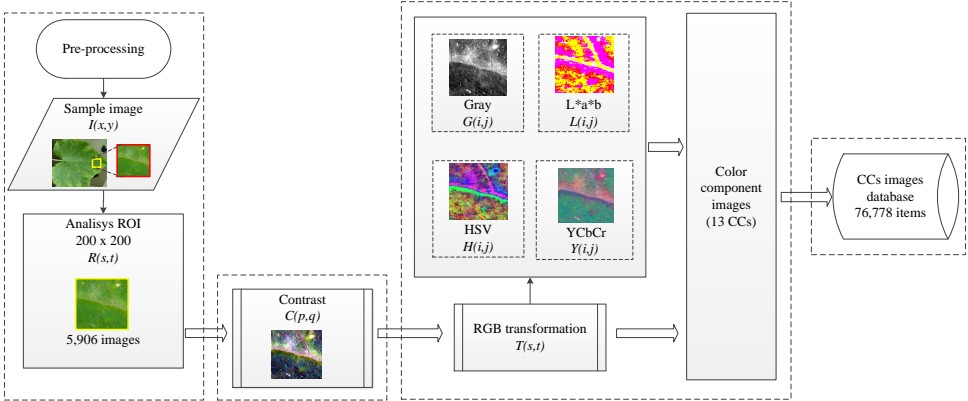

**Figure 5.** Preprocessing of the ROI images starting with the color transformation and separation of color components (CCs), where the sample image ($I(x, y)$) is the original image, which is followed by the analysis of ROI results in a new sample in RGB ($R(s, t)$), then a contrast adjust ($C(p, q)$) is performed to obtain the transformation of the image ($T(s, t)$) in the different color spaces ($G(i, j)$, $L(i, j)$, $H(i, j)$, $Y(i, j)$) and the separation for color components.

*2.4. Feature Extraction*

In this study, the color components of each space were analyzed to obtain relevant information. From the gray-level co-occurrence matrix (GLCM), texture features were extracted. The GLCM is a statistical method that takes into account the spatial relationship of pixels.

A GLCM matrix corresponds to a CC image considering the 255 gray levels, represented by the function $P(I, i, j, d, \theta)$, where *i* represents the gray level location $(x, y)$ in image $I(x, y)$, and *j* represents the gray level of the pixel at a distance $d = 1$ from the location $(x, y)$ with an orientation angle and normalized with Equation (1) [17].

$$p(i, j) = \frac{P(i, j, 1, 0)}{\sum_{i,j} P(i, j, 1, 0)} \tag{1}$$

Figure 6 shows the computation of a GLCM matrix of an image with gray intensity levels, where the neighboring pixel pairs could be matched with four different reference angles ($0°$, $45°$, $90°$, and $135°$). Figure 7 shows the GLCM matrices generated from the $H_i$ components in the HSV color space.

The TDs contain some information about shade, texture, shape, and color, describing the distribution, homogeneity, contrast, constant color, intensity, and gray levels of brightness. The TD equations based on the GLCM are presented in Table 1. An explanation of the feature extraction process used in our approach is given in Figure 8.

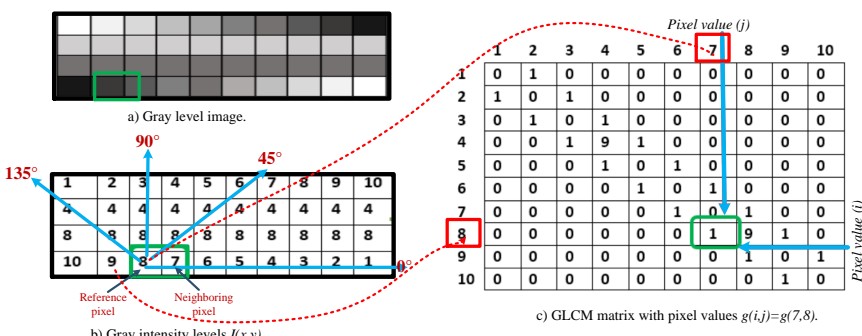

**Figure 6.** Calculation of the GLCM matrix in a gray image. The distance is $d = 1$, and the angle is $\theta = 0$: (**a**) gray image, (**b**) gray levels $I(x,y)$, and (**c**) GLCM matrix with the paired pixels $g(i,j)$.

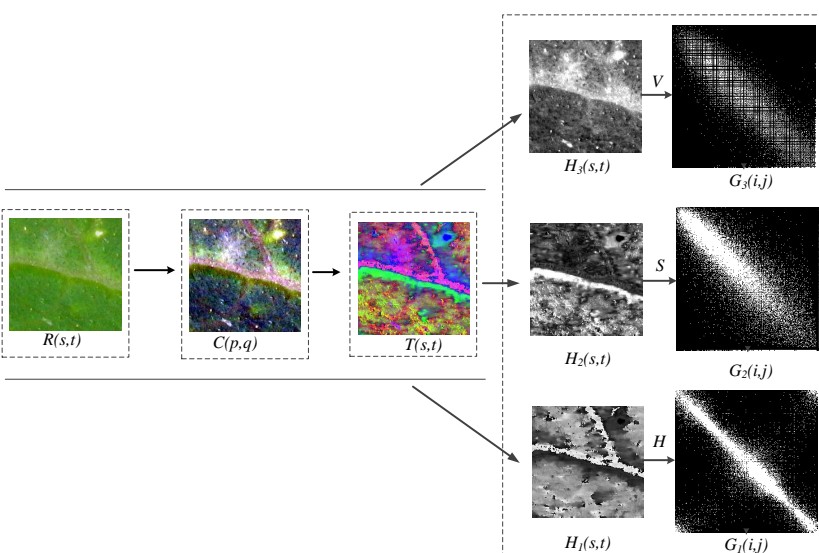

**Figure 7.** Processed image ($I(x,y)$ and $U(s,t)$); color transformation ($H(s,t)$); components $H_1(s,t)$, $H_2(s,t)$, and $H_3(s,t)$; and their GLCM matrices $G_1(i,j)$, $G_2(i,j)$, and $G_3(i,j)$ with 255 gray levels.

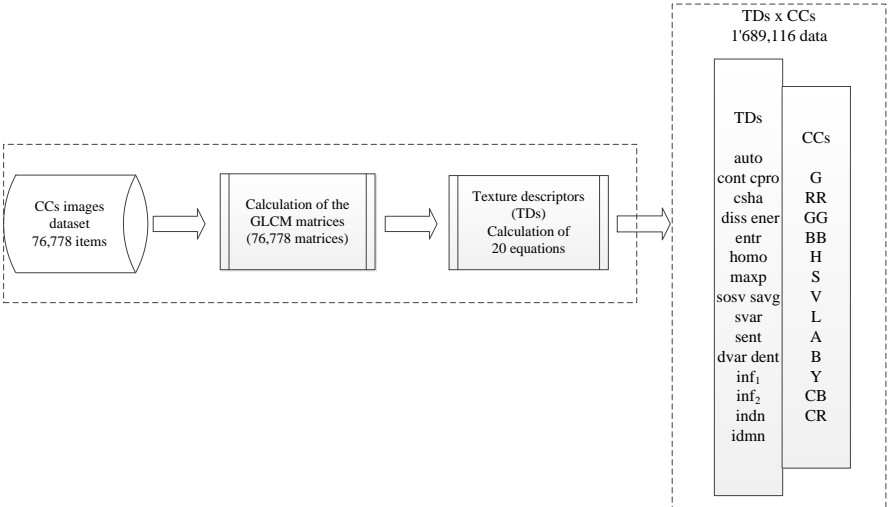

**Figure 8.** Process of feature extraction through the color component images.

**Table 1.** Texture descriptors (DTs) equations [4,17–19].

| Equation | DTs | Texture Descriptors |
|:---:|:---:|:---:|
| $\sum_{i,j}(i,j)p(i,j)$ | auto | Autocorrelation |
| $\sum_{i,j}\lVert i-j\rVert^2 p(i,j)$ | cont | Contrast |
| $\sum_{i,j}\frac{\{i\times j\}\times p(i,j)-\{\mu_x\times\mu_y\}}{\sigma_x\times\sigma_y}$ | corr | Correlation [1] |
| $\sum_{i,j}\{i+j-\mu_x-\mu_y\}^4\times p(i,j)$ | cpro | Cluster Prominence [1] |
| $\sum_{i,j}\{i+j-\mu_x-\mu_y\}^3\times p(i,j)$ | csha | Cluster Shade [1] |
| $\sum_{i,j}\lVert i-j\rVert\cdot p(i,j)$ | diss | Dissimilarity |
| $\sum_{i,j}p(i,j)^2$ | ener | Energy |
| $-\sum_{i,j}p(i,j)log_2(p(i,j))$ | entr | Entropy |
| $\sum_{i,j}\frac{1}{1-(i-j)^2}p(i,j)$ | homo | Homogeneity$_1$ |
| $max_{i,j}p(i,j)$ | maxp | Maximum Probability [1] |
| $\sum_{i,j}(i-\mu)^2 p(i,j)$ | sosv | Sum of Squares |
| $\sum_{i,j}ip_{x+y}(i)$ | savg | Sum Average |
| $\sum_{i,j}(i-j)2p(i,j)$ | svar | Sum Variance |
| $-\sum_{i,j}p_{x+y}(i)log(p_{x+y}(i))$ | sent | Sum Entropy |
| $\sum_{i,j}(k-\mu_x x-y)^2 p_{x-y}(k)$ | dvar | Difference Variance [1] |
| $-\sum_{i,j}p_{x+y}(i)log_2(p_{x+y}(i))$ | dent | Difference Entropy |
| $\frac{HXY-HXY_1}{max(HX,HY)}$ | inf$_1$ | Information Measure of Correlation$_1$ [2,3] |
| $\sqrt{1-exp[-2(HXY_2-HXY)]}$ | inf$_2$ | Information Measure of Correlation$_2$ [2,3] |
| $\sum_{i,j}\{i-j\}^2\times p(i,j)$ | indn | Inverse Difference Normalized |
| $\sum_{i,j}\frac{1}{1+(i-j)^2}p(i,j)$ | idmn | Inverse Difference Moment Normalized |

[1] $\mu_x, \mu_y$ and $\sigma_x$, and $\sigma_y$ are the median and standard deviation of $p_x$ and $p_y$, respectively. [2] $HXY$ = entr, where $HX$ and $HY$ are the entropies of $p_x$ and $p_y$, respectively. [3] $HXY_1 = -\sum_{i,j}p(i,j)log\{p_x(i)p_y(j)\}$ and $HXY_2 = -\sum_{i,j}p_x(i)p_y(j)log\{p_x(i)p_y(j)\}$.

A total of 260 features (20 TDs × 13 CCs) were extracted from 76,778 GLCM matrices from the Color Component Image Dataset (CC-ID), which generated a texture dataset of 1,535,560, labeled as abbreviated texture descriptor names followed by the color component: DTs-CCs. An example is the texture feature diss$_{BB}$ "dissimilarity" and the blue component (BB) of the RGB color space image. The texture dataset was normalized using the minimum and maximum values of each row.

## 2.5. Feature Selection

The feature selection process consists of finding the best set of features that allows us to differentiate the four levels of damage. Statistical methods are used for the selection process, the flow diagram of which is shown in detail in Figure 9 [16].

First, a Lilliefors test was performed to assess the normality of the trait data set, followed by an analysis of variance (ANOVA) to obtain statistical significance values, which was then followed by Tukey's test for multiple comparisons. The Lilliefors test compares the sample scores to a set of normally distributed scores with the same mean and standard deviation; the null hypothesis is that "the sample distribution is normal" [20–22]. Parameter values are "1" in the Lilliefors test with a determined $h$ value for each feature. Table 2 presents examples of three texture features (diss, homo, and idmn) in all component colors, where the calculated $h$ value = 1 means that the data have a normal distribution, and $h$-value = 0 means that they do not. As a result, we discarded 64 texture features.

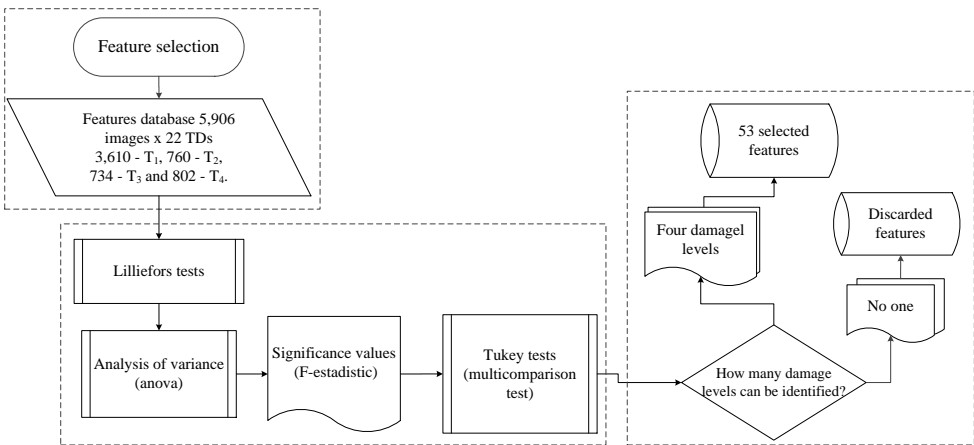

**Figure 9.** Feature selection process consists of a Lilliefors test, then an analysis of variance, and Tukey's test.

**Table 2.** An example of the results of two features submitted to the Lilliefors test. For each CC (G—gray, R—red, GG—green, BB—blue, H—hue, S—saturation, V—value, L—luminance, A—a * red and green coordinates, B—b * yellow coordinates with blue, Y—luma, CB—Cb chrominance difference of blue, CR—Cr chrominance difference of red). The features are shown with their four damage levels. If the *h*-value in " 0 " appears at any level, the feature is discarded for not complying with the normality condition.

| | | Gray | RGB | | | HSV | | | L * a * b | | | YCbCr | | |
|---|---|---|---|---|---|---|---|---|---|---|---|---|---|---|
| | TDs | G | R | GG | BB | H | S | V | L | A | B | Y | CB | CR |
| $T_1$ | diss | 1 | 1 | 1 | 1 | 1 | 1 | 1 | 1 | 1 | 1 | 1 | 1 | 1 |
| $T_2$ | | 1 | 1 | 1 | 1 | 1 | 1 | 1 | 1 | 1 | 1 | 1 | 1 | 0 |
| $T_3$ | | 1 | 1 | 1 | 0 | 1 | 1 | 1 | 1 | 1 | 1 | 1 | 1 | 0 |
| $T_4$ | | 0 | 1 | 0 | 0 | 1 | 1 | 0 | 1 | 1 | 1 | 0 | 1 | 1 |
| $T_1$ | homo | 1 | 1 | 1 | 1 | 1 | 1 | 1 | 1 | 1 | 1 | 1 | 1 | 1 |
| $T_2$ | | 1 | 1 | 1 | 1 | 1 | 1 | 1 | 1 | 1 | 1 | 1 | 1 | 1 |
| $T_3$ | | 1 | 1 | 1 | 1 | 1 | 1 | 1 | 1 | 1 | 1 | 1 | 1 | 1 |
| $T_4$ | | 1 | 1 | 1 | 1 | 1 | 1 | 1 | 1 | 1 | 1 | 1 | 1 | 1 |
| $T_1$ | idmn | 1 | 1 | 1 | 1 | 1 | 1 | 1 | 1 | 1 | 1 | 1 | 1 | 1 |
| $T_2$ | | 1 | 1 | 1 | 1 | 1 | 1 | 1 | 1 | 1 | 1 | 1 | 1 | 1 |
| $T_3$ | | 1 | 1 | 1 | 1 | 1 | 1 | 1 | 1 | 1 | 1 | 1 | 1 | 1 |
| $T_4$ | | 1 | 1 | 0 | 1 | 1 | 1 | 1 | 1 | 1 | 1 | 1 | 1 | 1 |

Analysis of variance (ANOVA) is a statistical method used to test for differences between two or more mean values. ANOVA is applied to test general rather than specific differences between mean values; we used ANOVA to test the null hypothesis $H_0$ in Equation (2) that the average values of the four PM damage levels ($T_1$, $T_2$, $T_3$, and $T_4$) are equal for each texture characteristic.

$$H_0 : \mu_{T_1} = \mu_{T_2} = \mu_{T_3} = \mu_{T_4} \tag{2}$$

The F statistic and $p < 0.000001$ were calculated for all texture characteristics. Following this, for each pair of information ($T_1$ versus $T_2$, $T_1$ versus $T_3$, $T_1$ versus $T_4$, $T_2$ versus $T_3$, $T_2$ versus $T_4$, and $T_3$ versus $T_4$), a multicomparison was performed with Tukey's test [20], labeled with different lowercase letters ("a", "b", "c" or "d"), and assigned when the comparison between the mean value of each damage level was different; otherwise, it was the same lowercase letter. If the same lowercase letter appeared in two, three, or four levels, there was no significant difference in their respective texture characteristic. Finally, 53 texture features had significantly different mean values among the four damage levels. Table 3 presents only 17 texture features that were more sensitive to the discrimination of

the four damage levels. Figure 10 shows the results of ANOVA and Tukey's test of the $diss_{BB}$ feature, which presents mean values with significant differences between the four damage levels, and the $auto_A$ feature, which could separate only $T_1$ from $T_2$, $T_3$, and $T_4$.

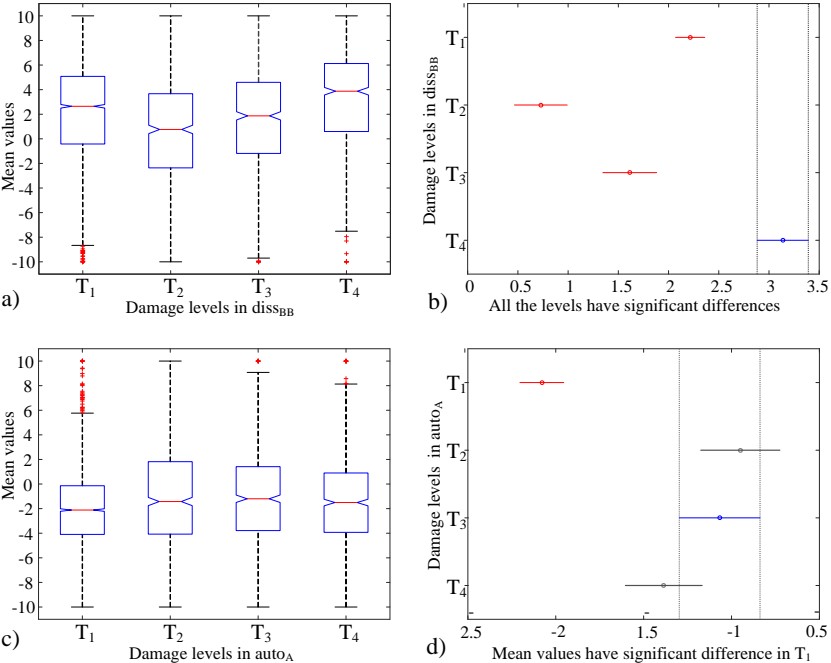

**Figure 10.** Results of the ANOVA and Tuke's test: (**a**) mean values of the damage levels of diss-BB; (**b**) Tukey's test, where the means of the damage levels are significantly different; (**c**) mean values of the damage levels of auto-A; (**d**) Tukey's test, where the means of $T_2$, $T_3$, and $T_4$ are equal but significantly different from $T_1$.

**Table 3.** Examples of results of Tukey's test by feature listed in order according to the ability to separate the four damage levels of PM.

| Feature | $F$ Statistic | $T_1$ | $T_2$ | $T_3$ | $T_4$ |
|---------|---------------|-------|-------|-------|-------|
| $ener_B$ | 184.7 | a | b | c | d |
| $corr_G$ | 174.7 | a | b | c | d |
| $homo_V$ | 171.2 | a | b | c | d |
| $corr_G$ | 158.6 | a | b | c | d |
| $ener_{GG}$ | 143.2 | a | b | c | d |
| $ener_V$ | 142.6 | a | b | c | d |
| $dent_A$ | 134.5 | a | b | c | d |
| $sosv_V$ | 71.4 | a | b | c | d |
| $dvar_A$ | 125.5 | a | b | c | d |
| $idmn_A$ | 124.4 | a | b | c | d |
| $cpro_{GG}$ | 122.4 | a | b | c | d |
| $homo_G$ | 119.4 | a | b | c | d |
| $entr_S$ | 112.7 | a | b | c | d |
| $homo_Y$ | 111.3 | a | b | c | d |
| $cont_L$ | 109.7 | a | b | c | d |
| $dvar_L$ | 109.7 | a | b | c | d |
| $dvar_{GG}$ | 105.9 | a | b | c | d |

## 2.6. Formation of the Feature Vectors

Ten feature vectors were created for the training, validation, and testing processes from the 53 features (TD) with a significant difference between the four classes (PM damage levels) considered. The features were listed in ascending order (significance value *F*) to form the first group of five vectors ($F_1, \ldots, F_5$) containing six TDs comprising the different color space characteristics. The second group of vectors ($G_1, \ldots, G_5$) contained the same

number of TDs and components from the same color space (Table 4). The characteristics matrix was 35,436 × 6 texture features × 4 damage levels.

**Table 4.** Formation of the features vectors with the combination of six TDs features belonging to different color spaces ($F_1, \ldots, F_5$), and the features belonging to components of the same color space ($G_1, \ldots, G_5$).

| TDs | Vector | Features |
|---|---|---|
| Different color space combinations | $F_1$ | $\text{auto}_V$, $\text{dent}_S$, $\text{svar}_V$, $\text{svag}_L$, $\text{sosv}_V$, $\text{savg}_G$ |
| | $F_2$ | $\text{entr}_R$, $\text{homo}_R$, $\text{idmn}_R$, $\text{idmn}_{CR}$, $\text{dvar}_R$, $\text{cont}_R$ |
| | $F_3$ | $\text{dvar}_{CR}$, $\text{cont}_{CR}$, $\text{idmn}_Y$, $\text{idmn}_G$, $\text{idmn}_{GG}$, $\text{cont}_Y$ |
| | $F_4$ | $\text{cont}_L$, $\text{homo}_Y$, $\text{entr}_S$, $\text{homo}_G$, $\text{cpro}_{GG}$, $\text{idmn}_A$ |
| | $F_5$ | $\text{cont}_A$, $\text{dvar}_A$, $\text{dent}_A$, $\text{ener}_V$, $\text{ener}_{GG}$, $\text{corr}_L$ |
| Same color space combinations | $G_1$ | $\text{sent}_R$, $\text{entr}_R$, $\text{idmn}_{GG}$, $\text{cont}_{GG}$, $\text{diss}_{BB}$, $\text{inf}_{1BB}$ |
| | $G_2$ | $\text{dent}_S$, $\text{entr}_S$, $\text{auto}_V$, $\text{svar}_V$, $\text{sosv}_V$, $\text{ener}_V$ |
| | $G_3$ | $\text{diss}_L$, $\text{savg}_L$, $\text{idmn}_A$, $\text{cont}_A$, $\text{dvar}_A$, $\text{ener}_B$ |
| | $G_4$ | $\text{diss}_Y$, $\text{homo}_Y$, $\text{corr}_Y$, $\text{idmn}_{CR}$, $\text{dvar}_{CR}$, $\text{cont}_{CR}$ |
| | $G_5$ | $\text{diss}_G$, $\text{savg}_G$, $\text{idmn}_G$, $\text{cont}_G$, $\text{dvar}_G$, $\text{homo}_G$ |

*2.7. Proposed Multiclass Classification Framework*

The main objectives of the proposed framework are to implement support vector machines (SVMs) to classify PM damage levels ($T_1$–$T_4$) and predict the early phase. In addition, a multiclass problem is found in multiple binary classification cases, called one-vs.-one, resulting in a class comparison between each class. A block multiclassifier with $k(k-1)/2$ binary classifiers (SVMs) was constructed, where $k$ is the class number. An SVM is trained with different kernels (polynomial, sigmoidal, and radially-based Gaussian functions) to find the optimal hyperplane [23]. Hyperplane minimizing and estimating $h$ are performed using $h_{est} = R^2 ||w||^2 + 1$, where $R$ is the diameter of the smallest sphere including all training data, and $||w||$ is the vector of standard Euclidean weights. Therefore, an SVM classifies correctly when the parameters $\Gamma$ (confidence interval) and $h_{est}$ working with different values are minimied. In this study, 60% of the data were used for the training and validation, and 40% were used for the test. Six binary SVMs ($M_1, \ldots, M_6$) were trained with their corresponding two different classes of input data for all the feature vectors defined above. Tables 5 and 6 show the results for the SVMs trained with feature vectors whose components belong to different color spaces $F_1, \ldots, F_5$, and the same color space $G_1, \ldots, G_5$, respectively. In both tables, $p$ is the degree of the polynomial, $\omega$ is the variable parameter for the sigmoidal function, and $\sigma$ is the parameter for the radial basis function. The selected SVMs were those with the minimum values. The 2D graphs and 3D hyperplanes; and training, validation, and error results with different kernels and feature vectors are described in Figures 11 and 12, respectively.

**Table 5.** Support vector machines $M_1, \ldots, M_6$ with different space color components $F_1, \ldots, F_5$ with the kernels' linear, polynomial, sigmoidal, and radial base function for the selection of the SVM.

| Kernel | SVM | $p/\omega/\sigma$ | $R^2$ | $h_{est}$ | $\Gamma$ | $||w||^2$ | % Error |
|---|---|---|---|---|---|---|---|
| Lineal | $M_1$ | - | 433.36 | $1.0 \times 10^{15}$ | $0.0 + 90.74$ | $2.4 \times 10^{12}$ | 17.4 |
| Lineal | $M_3$ | - | 448.28 | $1.1 \times 10^{15}$ | $0.0 + 9.53i$ | $2.6 \times 10^{12}$ | 18.6 |
| Polynomial | $M_4$ | 4 | 3917 | $1.2 \times 10^{16}$ | $0.0 + 3.0i$ | 3105.6 | 30 |
| Polynomial | $M_6$ | 4 | 3989 | $7.1 \times 10^{16}$ | $0.0 + 8.7i$ | 1801.79 | 20.2 |
| Sigmoidal | $M_1$ | 1 | 2192.8 | 1096.5 | $0.0 + 1.67i$ | 500 | 17.6 |
| Sigmoidal | $M_4$ | 7 | 1614.5 | 8072.3 | $0.0 + 9.1i$ | 500 | 43.4 |
| RBF | $M_1$ | 0.5 | 0.9978 | 460.17 | 4.1048 | 461.18 | 0 |
| RBF | $M_2$ | 0.5 | 0.9977 | 464.95 | 4.1237 | 465.98 | 0 |
| RBF | $M_4$ | 0.5 | 0.9979 | 493.85 | 4.2359 | 494.87 | 0 |
| RBF | $M_5$ | 0.5 | 0.9979 | 487.92 | 4.2132 | 488.94 | 0 |
| RBF | $M_3$ | 1 | 0.9972 | 413.45 | 3.9135 | 414.60 | 0 |
| RBF | $M_6$ | 1 | 0.9974 | 456.07 | 4.0885 | 457.22 | 0 |

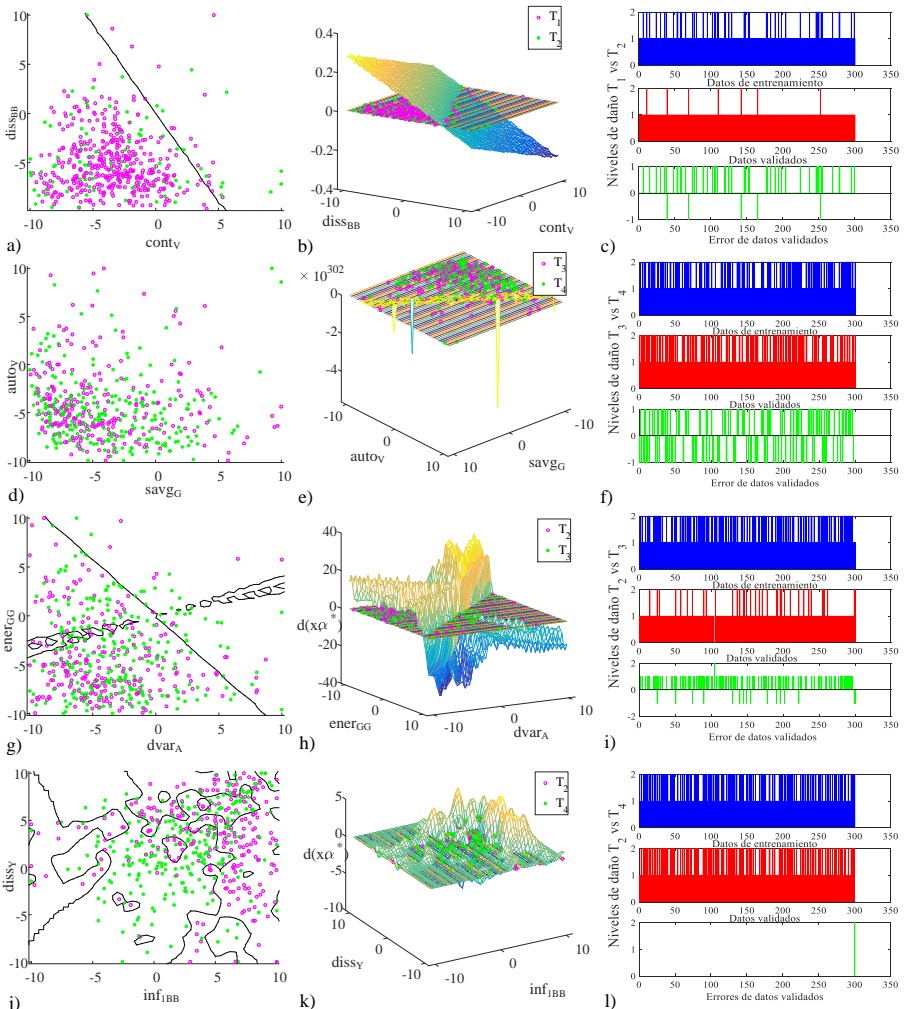

**Figure 11.** Kernel selection in the multiclassification system with the feature vectors in different color spaces with the optimal hyperplane: (**a**) linear kernel in 2D with $diss_{BB}$ versus $cont_V$, (**b**) 3D optimal hyperplane, (**c**) training and validation data with error in SVM $T_1$ versus $T_2$, (**d**) polynomial kernel in 2D with $auto_V$ versus $savg_G$, (**e**) 3D optimal hyperplane, (**f**) training and validation data with error in SVM $T_3$ versus $T_4$, (**g**) sigmoidal kernel in 2D with $ener_{GG}$ versus $dvar_A$, (**h**) 3D optimal hyperplane, (**i**) training and validation data with the error in SVM $T_2$ versus $T_3$, (**j**) radial base function kernel in 2D with $diss_Y$ versus $inf_{1BB}$ con kernel RBF, (**k**) 3D optimal hyperplane, and (**l**) training and validation data with the error in SVM $T_2$ versus $T_4$.

**Table 6.** Support vector machines with components of a space color $M_1, \ldots, M_6$ with the kernels' linear, polynomial, sigmoidal and radial base function for the selection of the SVM.

| Kernel | SVM | $p/\omega/\sigma$ | $R^2$ | $h_{est}$ | $\Gamma$ | $\|\|w\|\|^2$ | % Error |
|--------|-----|-------------------|-------|-----------|----------|---------------|---------|
| Lineal | $M_1$ | - | 478.89 | $1.1 \times 10^{15}$ | $0.0 + 9.74i$ | $2.4 \times 10^{12}$ | 16.8 |
| Lineal | $M_3$ | - | 455.57 | $1.0 \times 10^{15}$ | $0.0 + 8.59i$ | $2.2 \times 10^{12}$ | 15 |
| Polinomial | $M_1$ | 6 | $9.95 \times 10^{18}$ | $4.97 \times 10^{21}$ | $0.0 + 26i$ | 500 | 16 |
| Polinomial | $M_6$ | 6 | $9.28 \times 10^{18}$ | $1.24 \times 10^{15}$ | $0.0 + 98.2i$ | 0.0001 | 0 |
| Sigmoidal | $M_1$ | 3 | 2137.45 | 1065.1 | $0.0 + 11.8i$ | 500 | 17.2 |
| Sigmoidal | $M_2$ | 3 | 2104.09 | 1052.5 | $0.0 + 11.1i$ | 500 | 17.8 |
| RBF | $M_2$ | 1 | 0.9957 | 458.48 | 4.098 | 460.42 | 0 |
| RBF | $M_3$ | 0.5 | 0.9979 | 469.96 | 4.1434 | 470.95 | 0 |
| RBF | $M_4$ | 0.5 | 0.9978 | 491.79 | 4.2280 | 492.79 | 0 |
| RBF | $M_5$ | 0.5 | 0.9979 | 488.64 | 4.2160 | 489.64 | 0 |

**Table 6.** *Cont.*

| Kernel | SVM | $p/\omega/\sigma$ | $R^2$ | $h_{est}$ | $\Gamma$ | $||w||^2$ | %Error |
|--------|-----|-------------------|-------|-----------|----------|-----------|--------|
| RBF | $M_1$ | 2 | 0.9799 | 34,676.99 | 25.8 | 35,385.5 | 0 |
| RBF | $M_6$ | 1 | 0.9962 | 764.31 | 5.1414 | 767.17 | 0 |

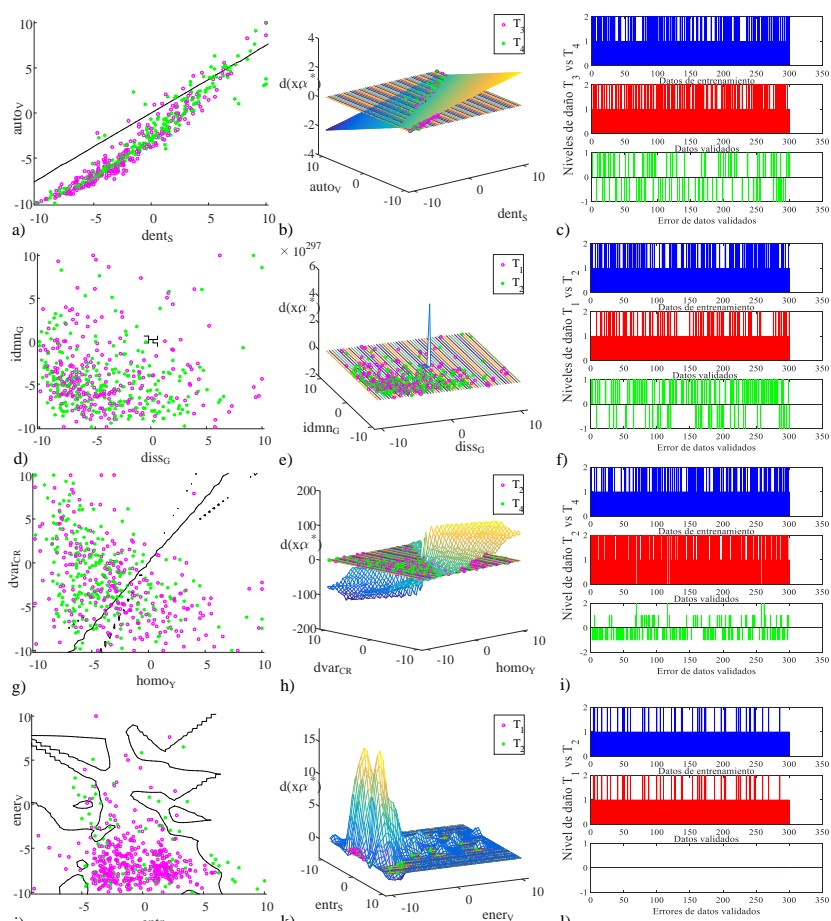

**Figure 12.** Kernel selection in the multiclassification system with the feature vectors in different color space with the optimal hyperplane: (**a**) radial base function kernel in 2D with $\text{auto}_V$ versus $\text{dent}_S$, (**b**) 3D optimal hyperplane, (**c**) training and validating data with the error in the SVM $T_3$ versus $T_4$, (**d**) linear kernel in 2D with $\text{idmn}_G$ versus $\text{diss}_G$, (**e**) 3D optimal hyperplane, (**f**) training and validate data with the error in the SVM $T_1$ versus $T_2$, (**g**) polynomial kernel in 2D with $\text{dvar}_{CR}$ versus $\text{homo}_Y$, (**h**) 3D optimal hyperplane, (**i**) training and validate data with the error in the SVM $T_2$ versus $T_4$, (**j**) radial base function kernel in 2D with $\text{ener}_V$ versus $\text{entr}_S$ con kernel RBF, (**k**) 3D optimal hyperplane, and, (**l**) training and validate data with the error in the SVM $T_1$ versus $T_2$.

The parameters for the best SVMs (different and same color spaces) were established using the one-versus-one (OVO) method for all class combinations ($T_1$ versus $T_2$, $T_1$ versus $T_3$, $T_1$ versus $T_4$, $T_2$ versus $T_3$, $T_2$ versus $T_4$, and $T_3$ versus $T_4$). Using a four-block voting scheme ($V_1, \ldots, V_4$), the final ranking decision of the assigned classes is made. When classes have the same number of votes, the one with the lowest index is selected. Figure 13 describes the OVO method and the voting scheme and classes that define each block. The best results of the testing stage are depicted in Figure 14 and Figure 15 for the features of different color spaces and the same color space, respectively.

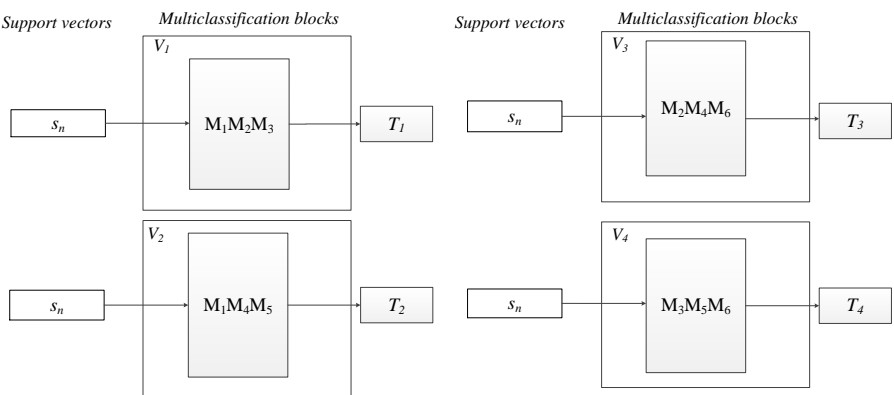

**Figure 13.** One-versus-one multiclassification method. The main inputs are the support vectors $s_1, \ldots, s_6$), the validation data for each binary classifier $M_1, \ldots, M_6$, and $\sigma$. Each block $V_1, \ldots, V_4$ contains the different support vector machines for multiple classification.

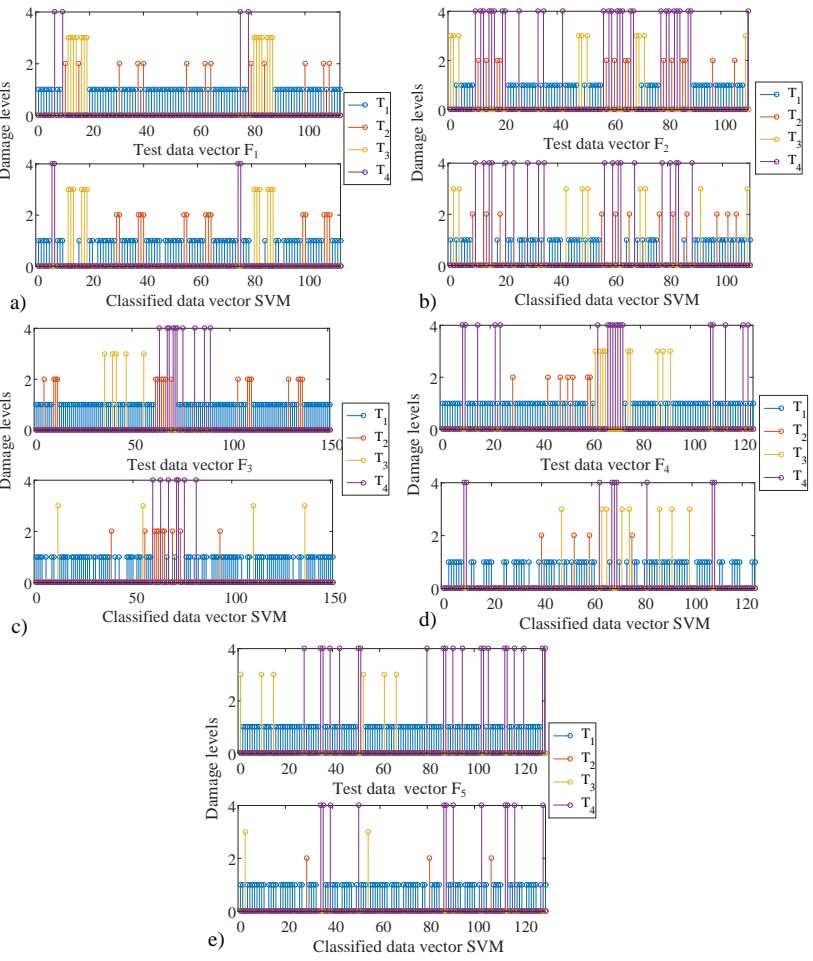

**Figure 14.** SVM binary classifiers: (**a**) test data $F_1$ and SVM-classified data, (**b**) test data $F_2$ and SVM-classified data, (**c**) test data $F_3$ and SVM-classified data, (**d**) test data $F_4$ and SVM-classified data, and (**e**) test data $F_5$ and SVM-classified data.

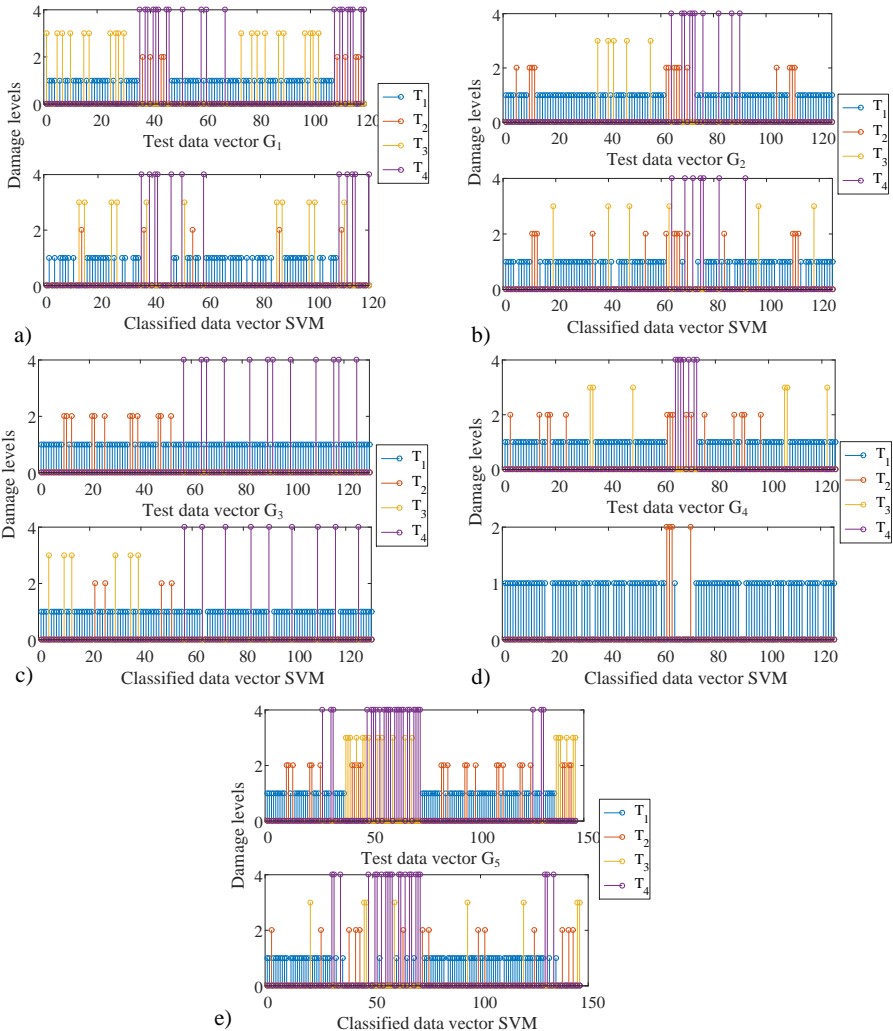

**Figure 15.** SVM binary classifiers with components of the same color space: (**a**) test data $G_1$ and SVM-classified data, (**b**) test data $G_2$ and SVM-classified data, (**c**) test data $G_3$ and SVM-classified data, (**d**) test data $G_4$ and SVM-classified data, and (**e**) test data $G_5$ and SVM-classified data.

*2.8. Performance Evaluation*

Metrics were calculated to evaluate the results. In this study, we used the confusion matrix, Cohen's kappa coefficients, accuracy, sensitivity, false positive range, F-statistic coefficients, and specificity to determine the performance of the proposed system. A confusion matrix allowed us to obtain the performance of the system in terms of the proportion of the total number of classified data: true positive ($TP$), which is the proportion of positive cases correctly identified; false positive ($FP$), which is the fraction of negative cases incorrectly classified as positive; true negative ($TN$), which is the proportion of negative samples correctly classified; and false negative ($FN$), which is the proportion of positive cases incorrectly distinguished [24–26]. The metrics were accuracy ($ACC$; Equation (3)), sensitivity ($SN$; Equation (4)), specificity ($SP$; Equation (5)), precision ($PRE$; Equation (6), false positive rate ($FPR$; Equation (7)), and F score ($F_\beta$; Equation (8)) [23,27].

$$ACC = \frac{TP + TN}{TP + TN + FN + FP} \qquad (3)$$

$$SN = \frac{TP}{TP + FN} = \frac{TP}{P} \qquad (4)$$

$$SP = \frac{TN}{TN + FP} = \frac{TN}{N} \qquad (5)$$

where $P$ is the positive total, and $N$ is the negative total.

$$PREC = \frac{TP}{TP + FP} \tag{6}$$

$$FPR = \frac{FP}{TN + FP} = 1 - SP \tag{7}$$

$$F_\beta = \frac{(1 + \beta^2)(PREC * SN)}{\beta2 * PREC + SN} \tag{8}$$

where natural values in $\beta$ are 0.5, 1, and 2, which was settto 1 in this case. Of the total of all the tests, the obtained metrics for system performance evaluation (percentages of $ACC$, $SN$, $SP$, $PREC$, $FPR$, and $F_\beta$) are shown in Tables 7 and 8. From the classified data from each test, we defined the confusion matrix ($n \times m$), where the rows ($n$) indicate the damage levels, and the columns ($m$) are the classes provided by the model. From this matrix, we can see when one class is confused with another. The diagonal components contain the sum of all the correct predictions, and the other diagonal components reflect the errors of the misclassified data [24,25]. Cohen's kappa coefficient (Equation (9)) is a statistical measurement of the interevaluator agreement for qualitative data or categorical variables. Its use in feature selection is suitable for testing the performance of models [28,29].

$$kappa = \frac{(d - q)}{(n - q)} \tag{9}$$

where $d$ is the sum of correctly classified data, and $q$ is the sum of each line and column in the confusion matrix to be divided by the total number of samples $n$ with kappa in [0–1], with concordance observed with degrees of agreements (between $k \geq 0$ and $k \leq 0.2$ is negligible, $k \geq 0.21$ and $k \leq 0.4$ is discreet, $k \geq 0.41$ and $k \leq 0.6$ is moderate, $k \geq 0.61$ and $k \leq 0.8$ is substantial, and $k \geq 0.81$ and $k \leq 1$ is perfect).

**Table 7.** Computed parameters for the performance evaluation of the classified data $F_1$, $F_2$, $F_3$, $F_4$, and $F_5$.

| Vector | $ACC(\%)$ | $SN$ | $SP$ | $PREC$ | $FPR$ | $F_\beta$ |
|--------|-----------|------|------|--------|-------|-----------|
| $F_1$ | 93.1 | 0.832 | 0.965 | 0.887 | 0.035 | 85.8 |
| $F_2$ | 88.4 | 0.700 | 0.945 | 0.811 | 0.055 | 75.1 |
| $F_3$ | 88.9 | 0.682 | 0.958 | 0.844 | 0.042 | 75.5 |
| $F_4$ | 90.0 | 0.728 | 0.957 | 0.850 | 0.043 | 78.4 |
| $F_5$ | 91.2 | 0.754 | 0.964 | 0.875 | 0.036 | 81.0 |

**Table 8.** Computed parameters for the performance evaluation of the classified data $G_1$, $G_2$, $G_3$, $G_4$, and $G_5$.

| Vector | $ACC$ | $SN$ | $SP$ | $PREC$ | $FPR$ | $F_\beta$ |
|--------|-------|------|------|--------|-------|-----------|
| $G_1$ | 87.3 | 0.625 | 0.956 | 0.824 | 0.044 | 0.711 |
| $G_2$ | 90.8 | 0.776 | 0.952 | 0.843 | 0.048 | 0.808 |
| $G_3$ | 94.4 | 0.877 | 0.967 | 0.898 | 0.033 | 0.887 |
| $G_4$ | 91.4 | 0.752 | 0.968 | 0.887 | 0.032 | 0.814 |
| $G_5$ | 87.3 | 0.678 | 0.938 | 0.786 | 0.062 | 0.728 |

## 3. Results

The extracted ROI images are proposed to obtain texture descriptors in three different color spaces and gray images. Next, we defined six feature vectors as a set with components from different color spaces and gray images. In addition, we detailed six feature vectors with color components from different color spaces and another six feature vectors with

the same color space. All these vectors contained the resulting 53 selected features. We used each feature vector into the trained multiclass SVM classifier, which categorized the input left image into four classes: healthy leaves, leaves with spore germination, leaves with the first symptoms, and diseased leaves: classes $T_1$, $T_2$, $T_3$, and $T_4$, respectively. A total of 130 samples of each class were used to test the system.

### 3.1. Different Color Space Feature Vectors

Table 9 shows the confusion matrices of the proposed early disease detection system for the feature vectors containing different color space characteristics. It shows the overall correctly classified and misclassified results of the defined disease levels for each feature vector.

Table 9 shows that the best success rate of the multiclass SVM classifiers with different color space feature vectors was 88.68% ($F_1$). Still, all the feature vectors $F_1$–$F_5$ could discriminate the four damage levels. For early detection, the optimal feature vector was $F_1$. Table 7 lists the performance evaluation results (Section 2.8), showing that the best feature vector was $F_1$ because this vector resulted in the best accuracy value of 93.1% and had a kappa = 0.7874, verifying the results in Table 10.

**Table 9.** Confusion matrix of the classified test data $F_1$–$F_5$.

| b) SVM-$F_1$ | $T_1$ | $T_2$ | $T_3$ | $T_4$ | Classified | % Correct |
|---|---|---|---|---|---|---|
| $T_1$ | 71 | 6 | 0 | 2 | 79 | 89.87 |
| $T_2$ | 2 | 9 | 0 | 0 | 11 | 81.82 |
| $T_3$ | 0 | 0 | 12 | 0 | 12 | 100.00 |
| $T_4$ | 2 | 0 | 0 | 2 | 4 | 50.00 |
| Test data | 75 | 15 | 12 | 4 | 106 | |
| % Correct | 94.67 | 60.00 | 100.00 | 50.00 | | **88.68** |
| d) SVM-$F_2$ | $T_1$ | $T_2$ | $T_3$ | $T_4$ | Classified | % Correct |
| $T_1$ | 46 | 5 | 2 | 2 | 55 | 83.64 |
| $T_2$ | 4 | 7 | 0 | 0 | 11 | 63.64 |
| $T_3$ | 3 | 0 | 7 | 0 | 10 | 70.00 |
| $T_4$ | 2 | 0 | 0 | 17 | 19 | 89.47 |
| Test data | 55 | 12 | 9 | 19 | 95 | |
| % Correct | 83.64 | 58.33 | 77.78 | 89.47 | | **81.05** |
| f) SVM-$F_3$ | $T_1$ | $T_2$ | $T_3$ | $T_4$ | Classified | % Correct |
| $T_1$ | 92 | 4 | 1 | 1 | 98 | 93.88 |
| $T_2$ | 4 | 5 | 3 | 0 | 12 | 41.67 |
| $T_3$ | 2 | 1 | 0 | 0 | 3 | 0.00 |
| $T_4$ | 3 | 0 | 0 | 6 | 9 | 66.67 |
| Test data | 101 | 10 | 4 | 7 | 122 | |
| % Correct | 91.09 | 50.00 | 0.00 | 85.71 | | **84.43** |
| b) SVM-$F_4$ | $T_1$ | $T_2$ | $T_3$ | $T_4$ | Classified | % Correct |
| $T_1$ | 74 | 4 | 2 | 1 | 81 | 91.36 |
| $T_2$ | 3 | 3 | 1 | 0 | 7 | 42.86 |
| $T_3$ | 1 | 1 | 6 | 0 | 8 | 75.00 |
| $T_4$ | 1 | 0 | 2 | 8 | 11 | 72.73 |
| Test data | 79 | 8 | 11 | 9 | 107 | |
| % Correct | 93.67 | 37.50 | 54.55 | 88.89 | | **85.05** |

**Table 9.** *Cont.*

| d) SVM-$F_5$ | $T_1$ | $T_2$ | $T_3$ | $T_4$ | Classified | % Correct |
|---|---|---|---|---|---|---|
| $T_1$ | 86 | 3 | 2 | 0 | 91 | 94.51 |
| $T_2$ | 0 | 0 | 0 | 0 | 0 | 0.00 |
| $T_3$ | 6 | 0 | 0 | 0 | 6 | 0.00 |
| $T_4$ | 3 | 0 | 0 | 12 | 15 | 80.00 |
| Test data | 95 | 3 | 2 | 12 | 112 | |
| % Correct | 90.53 | 0.00 | 0.00 | 100.00 | | **87.50** |
| Time | 5–6 ms | | | | | |

**Table 10.** Final performance evaluation.

| Vector | Features | *ACC* | **Kappa** | **% Correct** |
|---|---|---|---|---|
| $F_1$ | $\mathrm{auto}_V$, $\mathrm{dent}_S$, $\mathrm{svar}_V$, $\mathrm{savg}_L$, $\mathrm{sosv}_V$, $\mathrm{savg}_G$ | 0.931 | 0.7874 | 88.68 |
| $F_5$ | $\mathrm{cont}_A$, $\mathrm{dvar}_A$, $\mathrm{dent}_A$, $\mathrm{ener}_V$, $\mathrm{ener}_{GG}$, $\mathrm{corr}_L$ | 0.912 | 0.7841 | 87.50 |
| $G_3$ | $\mathrm{diss}_L$, $\mathrm{savg}_L$, $\mathrm{idmn}_A$ $\mathrm{cont}_A$, $\mathrm{dvar}_A$, $\mathrm{ener}_B$ | 0.944 | 0.7638 | 89.76 |
| $G_4$ | $\mathrm{diss}_Y$, $\mathrm{homo}_Y$, $\mathrm{corr}_Y$ $\mathrm{idmn}_{CR}$, $\mathrm{dvar}_{CR}$, $\mathrm{cont}_{CR}$ | 0.914 | 0.7835 | 88.68 |

*3.2. Same Color Space Feature Vectors*

Table 11 contains the confusion matrices of the proposed early-disease detection system for the feature vectors with the same color space characteristics. The best success rate achieved with the multiclass SVM classifiers with feature vectors using components of the same space color was 89.76% ($G_3$).

In feature vectors containing components of the same color space, higher accuracies were achieved in $G_3$ and $G_4$, with 94.4% and 91.4%, respectively (Table 8, in Section 2.8), with kappa = 0.7638 and kappa = 0.7835, respectively. These values are confirmed for the values presented in Table 11.

**Table 11.** Confusion matrix with the classified test data $G_1$, $G_2$, and $G_5$ in components of the same color space.

| b) SVM-$G_1$ | $T_1$ | $T_2$ | $T_3$ | $T_4$ | Classified | % Correct |
|---|---|---|---|---|---|---|
| $T_1$ | 56 | 3 | 2 | 1 | 62 | 90.32 |
| $T_2$ | 0 | 2 | 0 | 0 | 2 | 100.00 |
| $T_3$ | 6 | 0 | 6 | 0 | 12 | 50.00 |
| $T_4$ | 1 | 0 | 3 | 11 | 15 | 73.33 |
| Test data | 63 | 5 | 11 | 12 | 91 | |
| % Correct | 88.89 | 40.00 | 54.55 | 91.67 | | **82.42** |
| d) SVM-$G_2$ | $T_1$ | $T_2$ | $T_3$ | $T_4$ | Classified | % Correct |
| $T_1$ | 82 | 5 | 4 | 2 | 93 | 88.17 |
| $T_2$ | 2 | 9 | 1 | 0 | 12 | 75.00 |
| $T_3$ | 2 | 0 | 1 | 0 | 3 | 33.33 |
| $T_4$ | 2 | 0 | 0 | 5 | 7 | 71.43 |
| Test data | 88 | 14 | 6 | 7 | 115 | |
| % Correct | 93.18 | 64.29 | 16.67 | 71.43 | | **84.35** |

**Table 11.** *Cont.*

| b) SVM-$G_3$ | $T_1$ | $T_2$ | $T_3$ | $T_4$ | Classified | % Correct |
|---|---|---|---|---|---|---|
| $T_1$ | 101 | 0 | 2 | 0 | 103 | 98.06 |
| $T_2$ | 4 | 4 | 4 | 0 | 12 | 33.33 |
| $T_3$ | 0 | 0 | 0 | 0 | 0 | 0.00 |
| $T_4$ | 3 | 0 | 0 | 9 | 12 | 75.00 |
| Test data | 108 | 4 | 6 | 9 | 127 | |
| % Correcs | 93.52 | 100.00 | 0.00 | 100.00 | | **89.76** |
| d) SVM-$G_4$ | $T_1$ | $T_2$ | $T_3$ | $T_4$ | Classified | % Correct |
| $T_1$ | 90 | 0 | 0 | 0 | 90 | 100.00 |
| $T_2$ | 6 | 4 | 0 | 0 | 10 | 40.00 |
| $T_3$ | 4 | 0 | 0 | 0 | 4 | 0.00 |
| $T_4$ | 2 | 0 | 0 | 0 | 2 | 0.00 |
| Prueba | 102 | 4 | 0 | 0 | 106 | |
| % Corrects | 88.24 | 100.00 | 0.00 | 0.00 | | **88.68** |
| f) SVM-$G_5$ | $T_1$ | $T_2$ | $T_3$ | $T_4$ | Classified | % Corrects |
| $T_1$ | 67 | 3 | 0 | 2 | 72 | 93.06 |
| $T_2$ | 9 | 7 | 3 | 0 | 19 | 36.84 |
| $T_3$ | 3 | 2 | 5 | 0 | 10 | 50.00 |
| $T_4$ | 3 | 2 | 0 | 20 | 25 | 80.00 |
| Test data | 82 | 14 | 8 | 22 | 126 | |
| % Corrects | 81.71 | 50.00 | 62.50 | 90.91 | | **78.57** |
| Time | 5–6 ms | | | | | |

The final results and the features that are best for this diagnostic system are in Table 10, where the best feature vector is $G_3$. Deriving from the L * a * b color space produces the best component colors used for the identification according to the texture descriptors.

The features are combined according to the behavior of the pixels in the images. Gray-level co-ocurrence matrix textural properties such as contr, energy, corrm, corrp, dvarh, dissi, and idmnc, combined with color components such as R, GG, V, L, Y, and G, are accurate descriptors for diseased leaves versus healthy leaves and the intermediate damage levels, in general, without any specification required of the signs or symptoms. Our results are a textural analysis, which have the potential of being develoedp into a valuable evaluation tool that improves the diagnosis assessment of cucurbit plants. The features formed with contr and dvarh in the RGB, L * a * b, YCbCr, and gray color spaces are texture descriptors that describe significant differences among the four detected damage levels of powdery mildew. Contrast (contr) is a good feature for powdery mildew disease. This feature agrees with the texture characteristic, having high contrast values for large texture changes. The variance in statistics is a measurement that describes the spread between gray levels in an image. In our case, the variance difference (dvarh) measures how far the gray levels in the GLCM were from the mean value. Energy (energ) is a characteristic of the RGB color space in our GLCM that describes significant differences among the four damage levels. This measurement represents the local uniformity of the gray levels. It is an excellent descriptor for differentiating between white spots and infectious disease without uniformity in the samples regarding the first signs and symptoms. Also, energy is the angular second moment representing the uniformity in an image. We interpreted it as powdery mildew disease causing localized heterogeneity in a disease-specified area on the leaf, while the spores cause heterogeneous disorder throughout the whole leaf image. Contrast measures the quantity of local changes in an image. It reflects the sensitivity of the textures to changes in intensity. It returns a measure of the intensity contrast between a pixel and its neighborhood. Therefore, we considered high contrast as relevant for describing the signs he fungal disease. Contrast was 0 for a constant image in our samples that had a

lot of variation in color. C. pepo L. leaves have a local variation with consistently higher values. If a gray-scale difference occurs continually, the texture becomes coarse, and the contrast becomes large. Correlation is a descriptor that measures how correlated a pixel is to its neighborhood. It was used as a measure of the linear dependenciesof gray tone in our image samples. Feature values range from $-1$ to 1, defining a perfect negative and a positive correlation in the gray levels, respectively. The inverse difference moment normalized (idmnc) presents the difference between the neighboring intensity values that are normal by the total number of discrete intensity values. This means that in C. pepo L. leaves, all gray values of each damage level are considered according to 255 gray intensity values. The dissimilitude (dissi) in our samples showedthe variability between the gray levels that describe each damage level. For instance, a leaf with powdery mildew infection in an advanced state would present white spots with the green color disappearing. In cases of gray-color space, all the descriptors (corrp, corrm, homom, contr, dvarh, idmnc, and dissi) have a coincidence with the YCbCr space color in the Y and CR components.

## 4. Discussion

The present study provides a reference for the detection of PM damage levels in cucurbits. A feature dataset proved to be the best model for detecting four levels of PM damage on cucurbit leaves under natural crop conditions. As a result, the number of variables was reduced to minimize the calculation time. Using the images characterized with texture descriptors, it was possible to obtain a diagnosis using the leaves. The images describe the color changes visible on the leaves when symptoms are present. Therefore, the proposed texture descriptors are the result of calculations that integrate variables derived from the incidence of gray levels in the images of healthy and diseased leaves. Such is the case of a fungal disease that modifies pigmentation by affecting the photosynthesis process. Therefore, an image could describe this condition. Our main idea was to use a combination of color components and texture descriptors to detect infection through leaf color and texture. This study identified healthy and diseased leaves, defined as $T_1$ for healthy leaves and $T_4$ for diseased leaves, and two intermediate damage levels, $T_2$ for leaves with germinating spores and $T_3$ for leaves showing the first symptoms. Currently, to identify plant disease cases, methodologies are applied when symptoms are visible on leaves. However, early detection is the main problem and a main focus of studies on crops under field conditions. Thus, an advantage would be achieved by identifying the early stage $T_2$ of disease. The germinating spores on leaves are not visible and cannot be detected using a particular feature in image processing. As such, the implementation of these proposed features in future applications of sample classification for infected plants could lead to controlling the disease in time. A texture descriptor is a measurement that shows the heterogeneity in an image that is difficult to see with the human eye. Haralick textures [19] have been used for medical and biological research [18,30,31]. Haralick textures reveal the properties of the spatial distribution in a texture image. Computer diagnosis has been widely applied to characterize, quantify, and detect numerous plant situations such as the recognition of different leaves, medicinal plant classification, and detection of plant diseases and pests, for instance, in winter wheat, maize, citrus, and soybean [4,32]. Texture descriptors such as energy, entropy, contrast, homogeneity, and correlation have often been used in the literature [18,30–33]. These descriptors are measures that describe some visible features of leaves. The combination of these features changed when the leaf was diseased with PM, which indirectly modified the pixels in an image regarding the changes in the pigments. an infected leaf, the photosynthesis process slows, resulting in reductions in the chlorophyll and carotene contents. Then, the image contains different pixels with white or yellow spots according to the internal leaf conditions. In healthy young leaves, the green is lighter when they are younger. Mature leaves are darker green and have gray spots.

In some studies, image processing and other metrics have been used to identify differences in plant diseases. However, agreement on the damage levels of a fungal disease is still lacking. Researchers have focused on the discrimination of various diseases caused

by insects, viruses, and other pathologies. For cucurbit plants, some studies involved monitoring fungal or viral diseases with chemical analysis. Machine learning models have been used to distinguish plant diseases. According to a literature review, a limited number of researchers have studied disease damage level detection. To make the best decision for the control and monitoring of disease to enable protective measures to be implemented, the damage of plants over time must be measured using an optimal tool. In general, some methodologies can identify diseases, pests, viruses, and bacteria according to pathologies in different plants and crops. Image processing, spectroscopy, machine vision, remote monitoring, and hyperspectral imaging are tools that have been used for the identification of various visible symptoms and problems [34–36]. The white powdery mycelium covering the leaf surface affects the pixels in an image. The changes that occur on the leaf surface modify the pixels. A texture descriptor helps to identify gray levels based on disease damage. The proposed method calculates texture descriptors from ROI images of different spatial colors with various scales and crop databases. As shown in the results, the fungal disease of cucurbit plants affects the spectral signature of leaves in different ways, depending on the internal structure and disease characteristics.

An image displays these changes through the modification in gray levels. This study provides evidence that the analysis of textured features in an image simplifies the detection of a fungus based on the intensity of color feature data. We based the developed methodology on the most relevant combination of texture descriptors of a plant disease in combination with the health status, proposed damage levels, and stage classification, to identify healthy and diseased cucurbit plants.

Pydipati et al. [4] used the color co-occurrence method based on hue, saturation, and color intensity characteristics with uniformity, mean intensity, variance, correlation, product moment, inverse difference, entropy, and contrast using stepwise discriminant analysis to differentiate normal and diseased citrus leaves for diseases such as greasy spot, melanosis, and scab. They obtained identification accuracies between 100% and 95%. In this case, the diseases were detected when the symptoms were located on the leaves and were only compared with the characteristics of each pathogen. The backpropagation perceptron multilayer neural network performed classifications with descriptor textures in different medicinal plants, with accuracy values of between 75 and 80%. Ehsanirad et al. [32] developed a GLCM using texture descriptors such as autocorrelation, contrast, correlation, dissimilarity, energy, and entropy. In addition, they performed principal component analysis (PCA) for classification based on the leaf recognition of thirteen types of plants. The overall accuracy was 78.46%, and for PCA, the accuracy was 98.46%. Similarly, Malegori et al. [33] focused on the identification of biofilms and layers of microorganisms coated on a surface for materials such as steel, plastic, and ceramics through image analysis to define contaminated samples using defined texture and PCA descriptors. Similarly, for plant identification, Kadir et al. [31] implemented a Bayesian classifier in combination with shape, color, and texture features. On the flavia and foliage datasets, accuracies of 95% and 97.19% were achieved, respectively. However, in other work, texture descriptors were used in applications for the medical area to evaluate the sensitivity to Haralick texture features according to the gray levels used in the texture analysis: for glioma and prostate symptoms in patients by Brynolfsson et al. [18] and for the diagnosis of skin diseases such as allergic skin disorders, and viral, bacterial, and fungal skin diseases by Arabi et al. [35].

In our work, the analyses focused on identifying different pathologies when leaves are diseased and differences between the stage of disease using texture descriptors. Applying the concepts of texture descriptors, we developed an improved methodology with the ability to be used with image datasets with a deep approach that could provide the early detection of a fungal disease. All these methods have in common a classification process that is applied to different problems with crop plants in different environmental conditions. Several methodologies present high classification accuracies. Some of the processes are robust and complex. Nevertheless, they achieve the purpose of differentiating diseases. The main idea is to find different methods that are adapted to different crops to enable

prevention. Identifying plan damage in time is a complicated task, but with a discrimination method that measures damage status, early identification is possible, even if it is not visible. In our analyses, we tested methods to identify diseased leaves in outdoor crops. Future applications will explore the utility of these results in outdoor conditions to discriminate leaves. In this study, one of the limitations is the optical devices and experimental issues that affect the quality of the sample image. Depending on the environment and real conditions in an open field, in an image, the hue of objects may change with brightness. In real conditions, the sun, natural lighting, texture, distance from the camera, time, and weather can be influential factors that affect the image acquisition of plants. These are all parameters to consider for a sampling strategy that includes external conditions during data acquisition and problem identification. Therefore, the planned follow-up control will help identify leaves undergoing continuous changes during growth stages when a fungal disease or pathogen is present.

Two computational limits are noted: i) the image preprocessing time required for feature extraction according to the quality of the images and the proposed feature extraction method, and ii) the time required for the training and validation process used for future classification. To select a binary classifier, it is necessary to identify the behavior of the samples for a cross-validation process. The classification process depends on the number of samples and the portion of training data used in the implemented machine learning method. Different studies have worked on diseases with different features and methods. In our approach, we selected the features of images as the proposed optimal texture descriptors for cucurbit leaves. As a result, we obtained a high-accuracy performance with these features. Therefore, we consider our method useful for future studies and applications in plants with similar characteristics. With these results, some characterized color components converted into texture descriptors produced sufficient class separability to classify the proposed four PM damage levels and identify the fungal disease at an early stage. Finally, 53 extracted features could differentiate damage levels, according to the results of statistical tests. In this work, we implemented algorithms based on RGB sample images, a contrast algorithm, color transformation, and GLCM calculation to obtain a series of texture descriptors as features. We then used statistical analysis to reduce these features and evaluate the ability of the models to differentiate the damage levels of each feature. A feature dataset emerged as the best model for detecting four levels of PM damage in cucurbit leaves. As a result, we reduced the number of variables to reduce computational time. Nevertheless, we considered images with features under highly variable outdoor lighting conditions. This study tested a methodology for identifying diseased leaves in open-field growing conditions. Future applications will explore the utility of these results in outdoor conditions using the proposed method to analyze similar leaves with different damage in a data set.

## 5. Conclusions

The control of fungal disease and its ecological impact is expensive due to the need for a prevention point to reduce and optimize the applications of chemical treatments. The visual monitoring of a fungal disease is difficult because a diseased leaf may be mistaken for a healthy leaf due to the absence of visible symptoms during spore germination. First, we obtained a collection of images under open-field conditions of cucurbits. Second, we processed the images via contrast adjustment and color transformation. From the sample images, we then performed a feature extraction process using the texture descriptors of the sample images. Next, we employed statistical tools such as the Lilliefors test, one-way ANOVA, and Tukey's test to demonstrate the effectiveness of the method in assessing PM disease severity levels. Fifty-three texture descriptors from color components in the L*a*b, HSV, and YCbCr color spaces were found to be capable of showing potentially significant differences among e four PM damage levels. A sample dataset of the four cucurbit classes at different stages was used. This proposed methodology is suitable for disease detection for a variety of cucurbitaceous plants given the similarity in their growth stages and planting

areas. However, it is a subject requiring further deep analysis and study. Technologies such as machine learning, big data, and the Internet of Things (IoT) could help with the sampling and data collection phases given the wide range of varieties, environmental conditions, and number of samples and taking into account all the parameters involved such as climate, lighting, and optical devices. It could be laborious and limit such investigations. For other varieties and crops, the proposed method may not contribute to optimal detection, but it may contribute to a feasible comparison strategy for future implementations under field conditions to detect different diseases.

**Author Contributions:** Conceptualization, C.A.R.-R. and E.R.P.-H.; methodology, C.A.R.-R. and E.R.P.-H.; software, C.A.R.-R. and E.R.P.-H.; formal analysis, C.A.R.-R., E.R.P.-H., O.V.-C. and I.A.R.-P.; investigation, C.A.R.-R., E.R.P.-H., O.V.-C. and I.A.R.-P.; visualization, O.V.-C. and I.A.R.-P.; supervision, O.V.-C. and I.A.R.-P.; project administration, C.A.R.-R. and E.R.P.-H. All authors have read and agreed to the published version of the manuscript.

**Funding:** This research received no external funding.

**Data Availability Statement:** The data are unavailable due to privacy and ethical restrictions.

**Conflicts of Interest:** The authors declare no conflicts of interest.

## Abbreviations

The following abbreviations are used in this manuscript:

| | |
|---|---|
| PM | Powdery mildew |
| RGB | Red, green, and blue |
| HSV | Hue, saturation, and value |
| L*a*b | Luminance, red, and blue crominance |
| YCbCr | Luma component, Cb and Cr chroma components |
| ANOVA | Analysis of variance |
| ROI | Region of interest |
| GLCM | Gray-level co-ocurrence matrix |
| CC | Color component |
| TD | Texture descriptor |

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
