# Peer review of "Early-Stage Identification of Powdery Mildew Levels for Cucurbit Plants in Open-Field Conditions Based on Texture Descriptors"

_inventions, doi:10.3390/inventions9010008_

Round 1

Reviewer 1 Report

Comments and Suggestions for Authors

The paper underscores the pivotal role of continuous monitoring in thwarting diseases in field crops, notably fungal infections causing alterations in leaf pigments and subsequent yield losses. It advocates for proactive disease management through early detection methods. Presenting an innovative approach, the study utilizes RGB leaf images and a support vector machine (SVM) to discern powdery mildew in cucurbit plants.

Meticulously detailed, the paper meticulously outlines various methods employed, providing comprehensive insights into each step of the research process. Drawing from a comprehensive dataset spanning five growing seasons across diverse locations, the study employs twenty-two texture descriptors derived from the gray-level co-occurrence matrix (GLCM). These descriptors highlight the levels of damage, encompassing healthy leaves, germination time in leaves (indicative of fungal presence), initial symptoms, and diseased leaves.

Considering the comprehensive methodology, meticulous detailing of employed methods, and promising findings indicating a 94% prediction accuracy and Cohen's kappa value of 0.7638, I strongly advocate for accepting the paper in its current form. The thoroughness of research, coupled with the innovative approach for early disease detection in field crops, especially concerning powdery mildew in cucurbit plants, substantiates the paper's significance.

Author Response

Response to the Reviewer: Thank you for your appreciation and positive comments about our work. The English language was reviewed in the grammar and style.

Reviewer 2 Report

Comments and Suggestions for Authors

The authors have proposed a methodology of powdery mildew disease identification using RGB leaf images. The images are used to identify the damage levels. In the proposed method, the gray-level co-occurrence matrix (GLCM) is used to calculate 22 texture descriptors, which are then used as main features. The support vector machine (SVM) is used as a classifier. The authors conducted numerous experiments aimed at selecting the elements of the algorithms used that gave the best results. The effectiveness of the proposed method was also verified experimentally. The authors presented the results in the form of values of selected metrics.

The following issues should be addressed:
1. Please clearly state in the Introduction what is new and innovative in the presented method. How does the proposed approach differ with similar state-of-the-art approaches? Is it only a combination of well-known algorithms or does it bring something innovative, and some new techniques are proposed?
2. Can the proposed method also be used for other research problems?
3. Please evaluate the advantages and disadvantages of the proposed approach.
4. Under what conditions can the proposed method give worse results than other state-of-the-art approaches?

Comments on the Quality of English Language

The English language has to be improved when it to grammar and style.

Author Response

The authors have proposed a methodology of powdery mildew disease identification using RGB leaf images. The images are used to identify the damage levels. In the proposed method, the gray-level co-occurrence matrix (GLCM) is used to calculate 22 texture descriptors, which are then used as main features. The support vector machine (SVM) is used as a classifier. The authors conducted numerous experiments aimed at selecting the elements of the algorithms used that gave the best results. The effectiveness of the proposed method was also verified experimentally. The authors presented the results in the form of values of selected metrics.

Response to the Reviewer: Thank you for your appreciation and observations about our work.

The following issues should be addressed:

  1. Please clearly state in the Introduction what is new and innovative in the presented method. How does the proposed approach differ with similar state-of-the-art approaches? Is it only a combination of well-known algorithms or does it bring something innovative, and some new techniques are proposed?

Response: Thanks for your important comments. In the Introduction section, a new paragraph was added. This paragraph contains the enumerated innovations of our work.

  1. Can the proposed method also be used for other research problems?

Response: Thanks for your important observations and questions. This work presents an implemented methodology for crops. However, when the applications consist of data treatment, the authors consider that the proposed methodology for the feature extraction and the feature selection processes could be applied for some identification problems in which the sample is an image, and the research wants to the treatment of the information. In the Discussion section, in the eighth paragraph, the authors describe the possibility of using our methodology in similar plants with characteristics in terms of leaves and plants.

  1. Please evaluate the advantages and disadvantages of the proposed approach.

Response: Thanks for your important comments. About your questions about our work, one of the main advantages is that the proposed methodology is implemented in open field crops, with the Illumination conditions and the natural environment. According to our proposal, another main idea is a non-destructive methodology, which means, the sample leaves are in natural conditions in the plants, and in this case, the plants were protected, and the leaves were not cut off. About the disadvantages, two paragraphs in the Discussion section, the sixth and the seventh, respectively, talk about the limitations in technological and experimental terms.

  1. Under what conditions can the proposed method give worse results than other state-of-the-art approaches?

Response: Regarding this question, about the sampling process, the research must identify the real conditions of the crops and search the manner of the images to have the best resolutions and definitions of the characteristics of the plants. A conditioned environment in natural is needed to obtain the best image, according to the disease descriptions. In subsection 2.1 Acquisition is the description of the places and the environmental conditions of the crops.  In the data cases, firstly, is necessary to determine the data numbers and to review the data behavior in the classes. When the feature extraction process is finished, a new dataset is obtained, and then, is recommended to use the feature map representations to observe the data behavior, which helps to determine the machine learning methods for the classification. In the cases when the datasets are different, helps the application of statistical tests to verify the data with the statistical power and to improve the classification process. In addition, with this, is obtained the reduction of the data in the methodology to define a class.
